

# Chemical composition of soluble and insoluble particles around the last termination preserved in the Dome C ice core, inland Antarctica

Ikumi Oyabu[1,2], Yoshinori Iizuka[1], Eric Wolff[3], Margareta Hansson[4]

[1]Institute of Low Temperature Science, Hokkaido University, Sapporo 060-0819, Japan

[2] National Institute of Polar Research, Tokyo 190-8518, Japan

[3] Department of Earth Sciences, University of Cambridge, Cambridge CB2 3EQ, UK

[4] Department of Physical Geography, Stockholm University, Stockholm 106 91, Sweden

*Correspondence to*: Ikumi Oyabu (oyabu.ikumi@nipr.ac.jp)

**Abstract.** Knowing the chemical composition of particles preserved in polar ice sheets is useful for understanding past atmospheric chemistry. Recently, several studies have examined the chemical compositions of soluble salt particles preserved in ice cores from inland and peripheral regions in both Antarctica (Dome Fuji and Talos Dome) and Greenland (NEEM). On the other hand, there is no study that compares salt compositions between different sites in inland Antarctica. This study examines the chemical compositions of soluble salt particles around the last termination in the Dome C ice core,
and compares them to those from Dome Fuji. Particles larger than 0.45 μm are obtained from the ice core by an ice sublimation method, and their chemical compositions are analyzed using scanning electron microscopy and energy dispersive X-ray spectroscopy. The major soluble salt particles are $CaSO_4$, $Na_2SO_4$, and $NaCl$, which is the same as that from the Dome Fuji ice core. Time-series changes in the composition of these salts are similar to those for the Dome Fuji ice core. Specifically, from 25 to 18 ka, the ratio of $NaCl$ to $Na_2SO_4$ is variable, but generally the $CaSO_4$ and $NaCl$ fractions are high
and the $Na_2SO_4$ fraction is low. Between 18 and 17 ka, the $CaSO_4$ and $NaCl$ fractions decrease and the $Na_2SO_4$ fraction increases. Between 16 and 6.8 ka, the $CaSO_4$ and $NaCl$ fractions are low and $Na_2SO_4$ fraction is high. However, the sulfatization rate of $NaCl$ at Dome C is higher than that at Dome Fuji. We argue that this higher rate arises because at Dome C more $SO_4^{2-}$ is available for $NaCl$ to form $Na_2SO_4$ due to a lower concentration of $Ca^{2+}$.

## 1 Introduction

Paleoclimatic records up to 800 thousand years (kyr) have been reconstructed from deep ice cores drilled on the Antarctic ice sheets (e.g., Steig et al., 1998; Petit et al., 1999; Watanabe et al., 2003; EPICA, 2004, 2006). Glacial-interglacial changes are found in all those records, with cycles recurring roughly every 100 kyr in the longer ones (Petit et al., 1999; Watanabe et al., 2003; EPICA, 2004). However, regional differences in climate change occur due to different air-mass origins (Morgan et al., 2002). For example, based on ice cores, Pedro et al. (2011) found that the onset of the Antarctic Cold Reversal (ACR) was



different between Law Dome, a coastal site in the Indo-Pacific sector of the Southern Ocean region, and EPICA Dronning Maud Land (EDML) in the Atlantic sector of the Southern Ocean region. They suggested that such differences represent the influence of local or non-climatic signals at individual sites, and the finding underscores the need for caution in interpreting the phasing of interhemispheric climate changes from single-site records. As another example, Fischer et al. (2007) showed that non-sea-salt $Ca^{2+}$ flux, which is a proxy for terrestrial material, is about three times higher at EDML than at Dome C in the glacial period. The cause, they argued, was that EDML is much closer to South America and downwind of the cyclonically curved atmospheric pathway from Patagonia, which is the major source for $Ca^{2+}$ in the glacial period (Reijmer et al., 2002).

These examples illustrate the need to determine the spatial and temporal distributions of chemical compounds from aerosols. In particular, soluble aerosols have been examined using the ion species measured in the bulk by ion chromatography (e.g., Legrand et al., 1993, 1997) or by using fluorescence and absorbance methods (Röthlisberger et al., 2000a; Kaufmann et al., 2008; Bigler et al., 2011). Over the past decade, chemical compositions of individual soluble salt particles in Antarctic ice cores have been directly measured by use of a micro-Raman spectrometer with cryo-system (Ohno et al., 2005, 2006, 2013; Sakurai et al., 2010, 2011), and by a sublimation-EDS (energy dispersive X-ray spectroscopy) method (Iizuka et al., 2009, 2012a, 2012b, 2013; Oyabu et al., 2014, 2015).

Recently, Oyabu et al. (2014) described the temporal variation of the chemical composition of sulfate and chloride salts with several-hundred-years time resolution during the last termination in the Dome Fuji ice core, inland East Antarctica. They found that the dominant soluble salt particles are $CaSO_4$, $Na_2SO_4$, and NaCl. The source of $CaSO_4$ is direct emissions of gypsum as well as $CaCO_3$ that react in the atmosphere with $H_2SO_4$, whereas the $Na_2SO_4$ are secondary aerosols produced by sulfatization of NaCl. These two salts form as follows:

$$CaCO_3 + H_2SO_4 \rightarrow CaSO_4 + CO_2 + H_2O \text{ (Legrand et al., 1997) , and} \qquad \text{(R1)}$$

$$2NaCl + H_2SO_4 \rightarrow Na_2SO_4 + 2HCl \text{ (Legrand and Delmas, 1988)} \qquad \text{(R2).}$$

During the last termination, the major sulfate salt changed from $CaSO_4$ (in the last glacial maximum) to $Na_2SO_4$ at $17.3 \pm 0.4$ ka on the AICC2012 timescale. Moreover, NaCl particles were found in the early Holocene, which was not expected from previous studies (Oyabu et al., 2014).

At Talos Dome, a peripheral region of Antarctica, the chemical compositions of soluble salt particles of selected Last Glacial Maximum (LGM) and Holocene sections were also analysed by Iizuka et al. (2013). They found that the major soluble salt particles are also $CaSO_4$, $Na_2SO_4$, and NaCl, but reaction R2 was less frequent than that in the Dome Fuji ice core due to a reduced time of reaction for marine-sourced aerosol before reaching Talos Dome and/or to a reduced post-depositional effect from the higher accumulation rate at Talos Dome. In this way, the most abundant salt types are different between an inland and a peripheral region of Antarctica.

On the other hand, other than Dome Fuji, inland Antarctica has no other sites at which the soluble salt compositions have been analyzed. The Dome C ice core, drilled by the European Project for Ice Coring in Antarctica (EPICA), is one of the most studied Antarctic deep ice cores (EPICA, 2004; Wolff et al., 2006, 2010). During the last termination, the warming





starts around 18 ka, reaches a first maximum at 14.7 ka, followed by an interruption towards cooler conditions during the ACR. After that, the temperature increases again to reach the Holocene climate at 11.7 ka (Stenni et al., 2011). Similar trends are found in most Antarctic ice cores. However, the rate of warming during the last termination slows between 16.0 and 14.5 ka in the Atlantic sector that includes Dome Fuji, but not in the Indo-Pacific sector including Dome C (Stenni et al., 2011).

Such a difference should be found not only in temperature change but also in aerosol composition. To investigate how the aerosol composition compares between sites in inland Antarctica, we examine here the chemical composition of soluble salt particles in the Dome C ice core, focusing on the last termination.

## 2 Methods

### 2.1 Ice core samples

The Dome C ice core was drilled on the East Antarctic ice sheet (75°06' S, 123°21' E, 3223 m a.s.l.) from 1996 to 2004, supplying a climate record for the past 800 kyr (EPICA, 2004; Wolff et al., 2006, 2010). After being stored at the Laboratoire de Glaciologie et Géophysique de l'Environnement (France) cold storage facility, the core sections were moved to a cold laboratory at the Department of Physical Geography, Stockholm University (Sweden). There, they were preserved at temperatures of −25 °C. The sample depths used here are from 222.75 m (Middle Holocene: 6.8 ka) to 570.50 m (LGM:

26.3 ka) (Table 1). We selected 30 core bags, each containing a 55-cm-long core section, and then cut discrete samples from the top 0.10 m of each sections. Each discrete sample is a $10 \times 4 \times 3$ cm$^3$ cuboid.

### 2.2 Observation of non-volatile particles by the sublimation-EDS method

To analyse micron-sized particles preserved in the ice samples, we applied the sublimation-EDS method (Iizuka et al., 2009). The sublimation system used for this study is located at Stockholm University. We decontaminated the ice samples by

20 shaving about 1–2 mm off the surface using a clean ceramic knife, and pulverized one of the faces of the ice surface with a 5–7 cm depth range. This amount yields about 1-g of ice and corresponds to approximately two years for the Holocene (11.5–6.8 ka), three years for the termination (17.5–12.1 ka), and five years for the LGM (26.3–18.7 ka). The pulverized ice was then placed on a polycarbonate membrane filter in a sublimation chamber. Dry, clean air at -50 °C flowed through the chamber at a rate of 15 L min$^{-1}$ for 100 hours. The air was produced by an air compressor (Hitachi oil-free Bebicon, 0.75OP-

25 9.5GS5/6) at 0.55 Mpa, with an air dryer and a cleaner (SMC Coop. IDG60SV) such that the air contained no oil or solid particles exceeding 0.03 μm in diameter. During sublimation, the chemical compositions of particles are unaffected by acids in the samples, such as $H_2SO_4$, $HNO_3$, and HCl, and are unaffected by contamination from the sublimation system (Oyabu et al., 2014, 2015). After sublimation, each filter yielded more than 400 non-volatile particles exceeding 0.45-μm diameter. Particles on each filter were analysed at the Institute of Low Temperature Science, Hokkaido University, Japan using a JSM-

30 6360LV (JEOL) SEM (scanning electron microscope) and a JED2201 (JEOL) EDS (energy dispersive X-ray spectroscopy) system with 20-keV acceleration voltage. To be counted as a non-volatile particle, a particle had to contain at least one of Na,



Mg, Si, Al, S, Cl, K, and Ca, each with an atomic ratio (%) amount at least twice that of the error (%). Elements C, Cr, Fe, and Pt were also observed, but we interpret these peaks as artifacts from the membrane filter (C), sample mount (Cr), the stainless steel of the sublimation system (Fe), and filter coating (Pt). Other elements were only rarely detected. Of all the particles on a given filter, about 200 were chosen at random to analyse (Table 1). In total, 6108 particles from 30 ice samples

were analyzed. The mean diameter of all 6108 particles is $2.31 \pm 0.28$ μm.

From the elemental compositions of particles detected by EDS, the chemical compositions of single particles were determined. The non-volatile particles are classified into insoluble dust, soluble sulfate salts, and soluble chloride salts following Iizuka et al. (2009). Particles containing Si are assumed to be insoluble silicate minerals (dust), those containing S are assumed to have a sulfate salt, and those containing Cl are assumed to have chloride salts. More specifically, particles

containing Ca and S are assumed to have $CaSO_4$, whereas, those with Na and S have $Na_2SO_4$. Particles containing Na and Cl are assumed to have NaCl. Particles containing Ca and Cl are labeled as Ca-Cl, particles containing Ca without S and Cl are labeled as CaX, and particles containing Na without Si, S and Cl are labeled as NaX.

The mole numbers of $CaSO_4$, $Na_2SO_4$, NaCl, Ca-Cl, CaX, and NaX for each sample are calculated using the spectrum ratios of each element obtained by SEM-EDS. The detailed procedure for the calculation of moles follows that of previous studies

(Oyabu et al., 2014, 2015). In particular, the number of moles of Na, Ca, S, and Cl are calculated by assuming that each particle is a sphere whose radius is calculated from its cross-sectional area on the SEM image. Here we express the equivalent of X (i.e., the number of ions in the solution from species X), as [X]. If [Na] < [S], then $[Na_2SO_4]$ = [Na]. If [Na] > [S], then $[Na_2SO_4]$ = [S]. The same procedure applies to $[CaSO_4]$ and [NaCl]. As an example, when a sample contained Na and both Cl and S, it was calculated as containing both NaCl and $Na_2SO_4$ in a ratio determined by the calculated elemental

ratio of Cl to S in the EDS spectra. Uncertainty in the mole number was also calculated following Oyabu et al. (2014). We regard each particle as a sphere of revolution around the particle major axis, with major and minor axes taken from the particle image observed by SEM. The error of this assumption was obtained by picking 200 particles at random, then measuring the deviation between their actual shadow area and comparing to the idealized ellipse shadow area. As a result, 95% of the particles have areas that differed by less than 20%. The analysis error of SEM-EDS is also about 20%.

**2.3 Stable water isotope, ion and salt concentrations**

For each of the 30 bags, we used $Na^+$, $Ca^{2+}$, $Cl^-$, and $SO_4^{2-}$ concentrations from the 55-cm-long section-mean data, as well as the concentration from discrete samples at the top of each section. (See Table 2 for the data.) Ion concentrations were measured in five laboratories (Florence, Stockholm, British Antarctic Survey, Laboratoire de Glaciologie et Géophysique de l'Environnement and Copenhagen). Details are described in Littot et al. (2002).

We also used published datasets of the stable oxygen isotope and ion concentrations. The Dome C $\delta D$ records of 212 samples from 329.7 to 582.7 m were obtained from Jouzel et al. (2007). The ion concentrations ($Ca^{2+}$, $Na^+$, $SO_4^{2-}$, $Cl^-$, and $NO_3^-$) of 161 datapoints from 221.1 to 564.3 m were obtained from published data. The $Ca^{2+}$, $Na^+$, and $SO_4^{2-}$ concentrations were derived from Wolff et al. (2006), the $Cl^-$ concentration was derived from Röthlisberger et al. (2003), and the $NO_3^-$





concentration was derived from Röthlisberger et al. (2000b). Ages are given on the AICC2012 timescale (Veres et al., 2013). The ages of Dome Fuji records from Oyabu et al. (2014) are also given on the AICC2012 timescale using tie points provided by Fujita et al. (2015).

$Na_2SO_4$ and $CaSO_4$ concentrations are calculated using the $Ca^{2+}$, $Na^+$, $SO_4^{2-}$, $Cl^-$, and $NO_3^-$ concentrations by following Case IV of the ion-deduced method in Oyabu et al. (2014). This method calculates salt concentrations by assuming that $Ca^{2+}$ forms $CaSO_4$ prior to $Ca(NO_3)_2$ and before the $Na^+$ forms $Na_2SO_4$, whereas $Na^+$ forms $Na_2SO_4$ prior to NaCl. Present aerosol observations at Dome C show that the $Na^+$ concentration is lower than that of $SO_4^{2-}$ in summer, but higher in winter (Preunkert et al., 2008). This result implies that in winter NaCl likely travels to Dome C without sulfatization, even though the annual mean concentration of $SO_4^{2-}$ exceeds that of $Na^+$. Assuming that reaction R2 occurred completely within each single month, we calculated the monthly $Na_2SO_4$ and NaCl concentrations using the monthly concentrations of $Na^+$ and $SO_4^{2-}$ provided by Preunkert et al. (2008). As a result of integrating monthly NaCl and $Na_2SO_4$ concentrations, Oyabu et al. (2014) found that 23% of the $Na^+$ comes from NaCl. Therefore, the ion-deduced method Case IV employs the molar equivalent ratio of NaCl to $Na^+$ of 0.23 instead of the zero value in the case that the $SO_4^{2-}$ concentration exceeds that of $Na^+$. This case occurs after 16.0 ka here.

## 3 Results and Discussion

### 3.1 Elemental combination of non-volatile particles

We examined the number of silicate dust, chloride salts and sulfate salts. From Fig. 1, we find that 60.1% of all particles contain Si, and 68.9% of all particles contain soluble components. The chloride salts (Cl- and Si-Cl-particles) account for 5.6% of all particles, and about half of them coexist with Si. The sulfate salts (S- and Si-S-particles) account for 46.8% of all particles, and about half of them coexist with Si. Particles containing both chloride and sulfate salts (Cl-S- and Si-Cl-S-particles) account for 7.6% of all particles, and 71.1% of them coexist with Si. The chloride and sulfate salts together account for 87.1% of soluble particles. The number of sulfate salt particles is more than 4 times as large as that of chloride salt, therefore the sulfate salts are the major soluble salts in the Dome C ice core in the period 26.3–6.8 ka. The major cations in inland Antarctica are $Ca^{2+}$ and $Na^+$ (e.g., Legrand and Mayewski, 1997), making the major soluble salts here the soluble calcium salts (Ca-salts) and soluble sodium salts (Na-salts). This previous finding is consistent with the results in Fig. 2 showing that Ca-salts and Na-salts constitute 91.3% of all soluble salt particles. The most dominant Na-salt is $Na_2SO_4$ and the second most dominant is NaCl. For the Ca-salts, $CaSO_4$ dominates. Thus the major soluble salts are $Na_2SO_4$, $CaSO_4$, and NaCl in the Dome C ice core. This result agrees with that found from the Dome Fuji ice core during the last termination (Oyabu et al., 2014).



### 3.2 General trend of the time series variation of non-volatile particles

Consider the fraction of Si-containing particles (insoluble dust) to all non-volatile aerosols. Figure 3 shows a high ratio of $86.4 \pm 4.9$ % in the LGM, but the ratio starts to decrease at 18.7 ka, and reaches a minimum around 13 ka. After that, the ratio slightly increases, averaging to $48.1 \pm 10.5$% after 12.1 ka. Namely, more than half of the non-volatile aerosols consist

of soluble salt particles in this period. The dust concentration of Dome C (Delmonte et al., 2004a) shows a similar behavior, with the dust concentration decreasing dramatically at the beginning of the termination and reaching a low value in the middle of termination. These results indicate that the ratio of insoluble dust to non-volatile aerosols appears to change with the dust concentration. Compared to Dome Fuji, the trend in Fig. 3 is similar, but the values are lower. Specifically, Dome Fuji is $95 \pm 2$% in the LGM and $67 \pm 20$% in the Holocene (Oyabu et al., 2014). This result indicates that the fraction of

insoluble dust (to all non-volatile particles) in the Dome C ice core is lower than that in the Dome Fuji ice core.

Consider now the general trend in the mole fraction of Ca-salt and Na-salt over the same period. Figure 4a shows large variability in the LGM (26.3 to 18.0 ka) especially for NaCl and $Na_2SO_4$, but generally has relatively high $CaSO_4$ and NaCl fractions, and a relatively low $Na_2SO_4$ fraction. After 18.0 ka, the $CaSO_4$ and NaCl fractions decrease and that of $Na_2SO_4$ increases. In going from the LGM to the period of 17.5–11.7 ka, the average fraction for $CaSO_4$, changes from $32.9 \pm 17.9$ to

15 $10.7 \pm 6.6$%, whereas that for NaCl changes from $21.1 \pm 16.1$ to $10.5 \pm 7.6$%, and that for $Na_2SO_4$ changes from $38.1 \pm 19.9$ to $65.5 \pm 10.0$%.

We now examine the major sulfate salts. The ratio of $Na_2SO_4$ to $CaSO_4$ starts low at $1.46 \pm 0.84$ between 26.3 and 18.0 ka (Fig. 4b), then increases to $11.12 \pm 12.32$ through 6.8 ka. Namely, the $Na_2SO_4/CaSO_4$ ratio increases during the deglacial warming. For the major Na-salts, the ratio of NaCl to $Na_2SO_4$ is variable, but generally higher from 26.3–18.0 ka than from

20 17.5–6.8 ka. The average ratio is $0.93 \pm 1.02$ in the period 26.3–18.0 kyr and is $0.19 \pm 0.15$ between 17.5 and 6.8 ka (Fig. 4c). Overall, these trends in the $CaSO_4$, $Na_2SO_4$, and NaCl fractions are similar to those in Dome Fuji (Oyabu et al., 2014).

Following the ion-deduced method in Oyabu et al. (2014), $CaSO_4$ and $Na_2SO_4$ concentrations were calculated using $Ca^{2+}$, $Na^+$, $SO_4^{2-}$, $Cl^-$, and $NO_3^-$ concentrations. The resulting $CaSO_4$ concentration in Fig. 5 shows a high value of $1.24 \pm 0.28$ $\mu mol \ L^{-1}$ in the LGM, but decreases dramatically between 18.0 and 15.0 ka. After that, the concentration decreases

moderately, reaching the low value of $0.03 \pm 0.01$ $\mu mol \ L^{-1}$ around 11.0 ka. In the Holocene, the concentration stays low at around 0.03 $\mu mol \ L^{-1}$. The $Na_2SO_4$ concentration also starts high at $1.05 \pm 0.21$ $\mu mol \ L^{-1}$ in the LGM, and decreases after 16.3 ka. Between 15.0 and 12.5 ka (ACR), the $Na_2SO_4$ concentration plateaus at $0.61 \pm 0.06$ $\mu mol \ L^{-1}$. After the ACR, the $Na_2SO_4$ concentration decreases again, reaches a low value of $0.25 \pm 0.02$ $\mu mol \ L^{-1}$ at 11.5 ka. In the Holocene, the concentration stays low at around 0.3 $\mu mol \ L^{-1}$.

The results in these last two sections essentially agree with those found in the Dome Fuji ice core (Oyabu et al., 2014). The major soluble salts in Dome C are the same ($CaSO_4$, $Na_2SO_4$, and NaCl) as at Dome Fuji, and the pattern of changes during the last termination is also similar. Based on these results, we suggest that glacial–interglacial variations in compositions and





concentrations of soluble salts aerosols at the Dome C are also similar to those found by Iizuka et al. (2012b) at Dome Fuji. Differences in the details between the two sites are explored next.

### 3.3 Variability of the presence of NaCl and Na$_2$SO$_4$

Here, we examine causes of high NaCl and Na$_2$SO$_4$ variability. The cause may be inherent interannual variability of sea-salt

and sulfates. For example, the snow-pit measurements of Hoshina et al. (2014) in inland Antarctica found that the balance of Na$^+$, SO$_4^{2-}$, and Cl$^-$ concentrations vary over a several-year timescale, suggesting inherent variability. Although their measurements were ion measurements, our sublimation measurements of the particles covered just 5-cm of ice from each section, also representing several years of particle deposition. However, previous measurements of the mean ion concentrations (e.g., Röthlisberger et al., 2003) suggest little variability. Thus, we ask here whether the variability vanishes

when the ion concentrations are averaged over the entire section, and then address the chemical cause of the higher variability.

First, we examine the Na$^+$, SO$_4^{2-}$, and Cl$^-$ concentrations of the section mean and those of the top 5 cm of discrete sample. For our samples between 6.8 and 23.6 ka, the Cl$^-$ and Na$^+$ concentrations and ion ratios from the top 5 cm of each 55-cm section are similar to those of the section mean. In contrast, at 24.6 ka, for the section as a whole (553.30–553.80 m), we find

that the Cl$^-$/Na$^+$ ratio is close to the bulk seawater ratio of 1.165 (Table 2). However, if we take only the discrete sample at the top (553.30–553.35 m), then we find lower Cl$^-$ concentration and lower Cl$^-$/Na$^+$ ratio. Also, the SO$_4^{2-}$ is high in this sample. Thus, it is possible to understand a low ratio of NaCl/Na$_2$SO$_4$ at 24.6 ka, as observed in our sublimation data. However for other sections, the mean agrees with that in the top 5 cm.

Then why is there greater variability between 18 and 26 ka? During this period, the Cl$^-$/Na$^+$ ratio lies close to the bulk

seawater ratio. In this case, if we follow bulk chemistry (e.g., Röthlisberger et al., 2003), all the Na$^+$ should be allocated to NaCl, with no space for Na$_2$SO$_4$. However, if we use the ion-deduced method of Oyabu et al. (2014), the deduced NaCl/Na$_2$SO$_4$ ratio is often close to the sublimation NaCl/Na$_2$SO$_4$ ratio except for 23.3 and 26.3 ka (Table 2). This method takes into account not only Na$^+$ and Cl$^-$ concentrations but also Ca$^{2+}$ and SO$_4^{2-}$ concentrations. If Ca$^{2+}$ is relatively low compared to Na$^+$ and SO$_4^{2-}$, or SO$_4^{2-}$ concentration is relatively high compared to Ca$^{2+}$ and Na$^+$ concentrations, Na$_2$SO$_4$ is

produced whether Cl$^-$/Na$^+$ is close to sea-water ratio or not. These considerations indicate that a cause of the large variabilities in NaCl/Na$_2$SO$_4$ is the different sulfatization rates of NaCl. It is implied that even if Cl$^-$/Na$^+$ is close to the seawater ratio, Na$^+$ may not simply be present as NaCl. Formation of Na$^+$ is changed by the presence of the other ions. We assume that Na$^+$ is transported and deposited as NaCl when the NaCl/Na$_2$SO$_4$ ratio is high, such as at 25.1 ka. On the other hand, when the NaCl/Na$_2$SO$_4$ ratio is low despite the Cl$^-$/Na$^+$ is close to the bulk seawater ratio, we assume that either (i) the

Na$^+$ is mainly deposited as NaCl, but has reacted after deposition, forming some Na$_2$SO$_4$ and also HCl, but the HCl remained in the snow, or (ii) the NaCl had reacted in the atmosphere forming Na$_2$SO$_4$ and HCl, but these products have been deposited in just the right ratio to give the seawater ratio of Cl$^-$/Na$^+$. Of these possibilities, (i) is more reasonable because if Na$_2$SO$_4$



was produced in the atmosphere, then Cl$^-$/Na$^+$ should have more variability because HCl is volatile and would not be expected to be transported with the aerosol.

Concerning the period after 17 ka, the ratio of NaCl to Na$_2$SO$_4$ stays low; a result that agrees with Röthlisberger et al.'s (2003) finding and also with our deduced value. However, the assumption that 23% of the Na$^+$ was present as NaCl is

sometimes an overestimate. It seems that NaCl/Na$^+$ fraction should lie between 0% (as implied by previous studies such as Röthlisberger et al. (2003) and Iizuka et al. (2008)) and 23% (proposed by Oyabu et al. (2014)). Namely, both estimations are not so unrealistic. In this way, most of the Na$^+$ was in the form of Na$_2$SO$_4$ after 17 ka; however, as Oyabu et al. (2014) suggested, some of the Na$^+$ transported as NaCl due to seasonality. Generally, NaCl usually remains without sulfatization during the LGM, whereas most of NaCl sulfatized in the Holocene, however, this is not always the case.

Finally, consider the cases at 23.3 and 26.3 ka, where the sublimation method shows very few NaCl particles and a near-zero NaCl/Na$_2$SO$_4$ ratio. As this more-accurate method disagrees with the ion-deduced value, it appears that NaCl sulfatization occurred that cannot be predicted by only using ionic balance. Thus, although the fraction of NaCl and Na$_2$SO$_4$ can be approximated using the ion-deduced method, a more precise fraction can be obtained only using the sublimation-EDS method.

**3.4 Differences of CaSO$_4$, Na$_2$SO$_4$, and NaCl fractions between Dome C and Dome Fuji**

Although the trends in salt fractions here are similar to those at Dome Fuji, there are differences that deserve further examination. The sulfatization rate of NaCl appears higher in Dome C than that in Dome Fuji. For evidence, consider Fig. 4b and 4c. The average ratio of Na$_2$SO$_4$/CaSO$_4$ at Dome C is 1.46 between 26 and 18 ka and thereafter is 11.12, whereas the same ratio in Dome Fuji is 0.66 between 25 and 18 ka and 5.05 between 16 and 11 ka. Thus, through the last termination, the

Na$_2$SO$_4$/CaSO$_4$ ratio is higher at Dome C than at Dome Fuji. For the NaCl/Na$_2$SO$_4$, the average ratio in Dome C is 0.93 between 26 and 18 ka and thereafter is 0.19. The same ratio in Dome Fuji is 2.77 between 25 and 18 ka and 0.54 between 16 and 11 ka. Thus, the NaCl/Na$_2$SO$_4$ ratio is lower at Dome C than at Dome Fuji. Combining these comparisons, the ratio of Na$_2$SO$_4$ against CaSO$_4$ and NaCl are both higher at Dome C than Dome Fuji, which means that greater NaCl sulfatization occurred at Dome C during the last termination. Also, there is almost no NaCl after 17.5 ka in Dome C, whereas

NaCl/Na$_2$SO$_4$ >1 until 16.9 ka in Dome Fuji. Thus, the timing when most of the NaCl sulfatized to Na$_2$SO$_4$ is about 600 years earlier at Dome C than at Dome Fuji.

Now compare the ion-deduced CaSO$_4$ and Na$_2$SO$_4$ concentrations. Over the entire period, the CaSO$_4$ concentration is lower at Dome C than at Dome Fuji (Fig. 5b, Table 3), whereas for 25–18 ka the Na$_2$SO$_4$ concentration is higher at Dome C than at Dome Fuji (Fig. 5c, Table 3). For the ion concentrations, the Ca$^{2+}$ concentration at Dome C is lower than that at Dome Fuji,

whereas the Na$^+$ and SO$_4^{2-}$ concentrations nearly agree for the two sites in the LGM and early Holocene (Table 3). According to previous studies, sulfatization of CaCO$_3$ is prior to that of NaCl in Antarctica (Röthlisberger et al., 2003; Iizuka et al., 2008; Sakurai et al., 2009; Oyabu et al., 2014). Given that the Ca$^{2+}$ concentration is lower than the SO$_4^{2-}$ concentration, if we assume that all of the Ca$^{2+}$ forms CaSO$_4$, then the CaSO$_4$ concentration in Dome C should be lower than that at Dome



Fuji because the $Ca^{2+}$ concentration at Dome C is lower than that at Dome Fuji. Then, more $SO_4^{2-}$ can form $Na_2SO_4$ at Dome C than at Dome Fuji. This argument indicates that NaCl sulfatization occurred more at Dome C than at Dome Fuji.

Due to the lower $Ca^{2+}$ concentration in Dome C, a different sulfatization rate of NaCl occurred. Consider the ion sources. At both locations, the major sources of $Na^+$ and $SO_4^{2-}$ are sea-salt and biogenic activity in the Southern Ocean, respectively (e.g., Wolff et al., 2010). Although the two locations face different sectors (Dome C: Indo-Pacific; Dome Fuji: Atlantic), the similar behaviour of sea-salt and biogenic activity inferred from the ion concentrations means the environment in the Southern Ocean was relatively uniform, suggesting that both sites reflect a large area of the Southern Ocean. On the other hand, the $Ca^{2+}$ concentration was clearly different. Fischer et al. (2007) showed that the non-sea-salt $Ca^{2+}$ flux at EDML is about three times higher than that at Dome C in the glacial period. They argued that the high level is due to the geographic location of EDML being much closer to South America and downwind of the cyclonically curved atmospheric pathway from Patagonia, which is the major source of $Ca^{2+}$ in this period (Reijmer et al., 2002). The same idea may be applied to explain a different $Ca^{2+}$ concentration between Dome C and Dome Fuji. In particular, Dome Fuji is much closer to Patagonia than Dome C, and some air parcels may come from the Patagonia region (Suzuki et al., 2008). Also, Delmonte et al. (2004b) examined the size distribution of dust from ice cores in three locations, and suggested that they involved significantly different dust-transport pathways during the last termination. Thus, the difference of site location between Dome Fuji and Dome C is a possible cause for the higher $Ca^{2+}$ concentration at Dome Fuji, which may have produced a different sulfatization rate of NaCl.

## 4 Conclusion

This study presented the chemical compositions of non-volatile particles around the last termination in the Dome C ice core by using the sublimation-EDS method. The major soluble salt particles were $CaSO_4$, $Na_2SO_4$, and NaCl, the same as that found previously in the Dome Fuji ice core. The period from 26.3 to 18.0 ka (LGM) is characterized by high fractions of $Na_2SO_4$ (38.1 ± 19.9%) and $CaSO_4$ (32.9 ± 17.9%), but a low fraction of NaCl (21.1 ± 16.1%). In contrast, the period of 17.5–11.7 ka is characterized by lower $CaSO_4$ (10.7 ± 6.6%), lower NaCl (10.5 ± 7.6%), and higher $Na_2SO_4$ (65.5 ± 10.0%). This trend is the same as that at Dome Fuji.

Basically, the NaCl fraction was high, but the $NaCl/Na_2SO_4$ ratio had high variability in the LGM. Sometimes, the $NaCl/Na_2SO_4$ ratio was nearly 0 despite $Cl^-/Na^+$ being close to the bulk seawater ratio. This result implies that $Na^+$ is not always in the form of NaCl, even when the $Cl^-/Na^+$ is close to the bulk seawater ratio. A cause of the low $NaCl/Na_2SO_4$ ratio could be that the $Na^+$ is mainly deposited as NaCl, but has reacted after deposition so that $Na_2SO_4$ has been formed, and also HCl, but the HCl has been retained quantitatively. After 17 ka, most of the $Na^+$ was in the form of $Na_2SO_4$, with just a small amount of NaCl. Generally, NaCl tends to remain without sulfatization during most of the LGM, and most of the NaCl sulfatized in the Holocene, which can also be estimated from ionic balance.



Although Dome C and Dome Fuji had similar trends, some differences occurred. For example, the ratio of $Na_2SO_4$ to both $CaSO_4$ and NaCl is higher at Dome C than Dome Fuji, indicating greater NaCl sulfatization at Dome C during the last termination. Moreover, Dome C has both a lower $CaSO_4$ concentration and a higher $Na_2SO_4$ concentration than that at Dome Fuji. The $Na^+$ and $SO_4^{2-}$ concentrations were almost the same at the two sites, whereas the $Ca^{2+}$ concentration is lower at

Dome C than at Dome Fuji. More $SO_4^{2-}$ was available for NaCl to form $Na_2SO_4$, and thus the sulfatization rate of NaCl was higher at Dome C than at Dome Fuji. The two sites had similar concentrations of $SO_4^{2-}$ and $Na^+$, which suggest either that the environment in the Southern Ocean was uniform across sectors, or that both sites are affected by a large area of the Southern Ocean. The lower $Ca^{2+}$ concentration at Dome C occurs because of the difference in distance of the site location to southern South America, the nearest major dust source.

**Acknowledgement**

We thank the logistics and drilling teams, responsible for the recovery of the Dome C ice core. This work is a contribution to EPICA, a joint European Science Foundation/European Commission (EC) scientific programme, funded by the EC (EPICA-MIS) and by national contributions from Belgium, Denmark, France, Germany, Italy, The Netherlands, Norway, Sweden, Switzerland and the UK. We are grateful to persons of Florence, Stockholm, British Antarctic Survey, Laboratoire de

Glaciologie et Géophysique de l'Environnement and Copenhagen who analysed ion concentrations, T. Karlin of Stockholm University for helping us to set up the sublimation system at Stockholm University, G. Teste of Laboratoire de Glaciologie et Géophysique de l'Environnemt for preparing ice samples, M. Furusaki for help with SEM-EDS analysis, and J. Nelson for help with revising the manuscript. This study was supported by Nordic Centre of Excellence Cryosphere-atmosphere interactions in a changing Arctic climate (CRAICC), by JSPS KAKENHI Grant Numbers 40370043, 26257201, 26610147,

and by the Grant for Joint Research Program of the Institute of Low Temperature Science, Hokkaido University. This work was also supported by the JSPS Institutional Program for Young Researcher Overseas Visits. The production of this paper was supported by a National Institute of Polar Research publication subsidy. E. Wolff is funded by the Royal Society. Inquiries about the data used in the study can be made to the authors.

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

| Depth (m) | Age (ka BP1950) | Number of particles | Average diameter |
|---|---|---|---|
| 222.75 | 6.8 | 205 | 2.13 |
| 233.75 | 7.2 | 212 | 2.00 |
| 241.45 | 7.5 | 207 | 2.59 |
| 251.35 | 7.9 | 201 | 2.35 |
| 276.65 | 8.9 | 197 | 2.49 |
| 287.10 | 9.3 | 200 | 2.38 |
| 296.45 | 9.6 | 206 | 2.32 |
| 305.25 | 9.9 | 206 | 3.09 |
| 313.50 | 10.2 | 210 | 2.45 |
| 321.75 | 10.5 | 204 | 2.56 |
| 334.95 | 10.9 | 202 | 1.91 |
| 350.35 | 11.5 | 202 | 2.34 |
| 368.50 | 12.1 | 202 | 2.44 |
| 377.30 | 12.5 | 207 | 2.02 |
| 386.65 | 13.0 | 194 | 2.33 |
| 395.45 | 13.5 | 205 | 2.05 |
| 403.15 | 13.9 | 201 | 2.61 |
| 422.95 | 14.9 | 202 | 2.38 |
| 430.65 | 15.3 | 202 | 1.98 |
| 449.35 | 16.4 | 210 | 1.98 |
| 458.15 | 16.9 | 199 | 2.53 |
| 465.85 | 17.5 | 206 | 2.18 |
| 481.25 | 18.7 | 204 | 2.56 |
| 505.45 | 20.6 | 202 | 2.18 |
| 521.95 | 21.9 | 205 | 2.13 |
| 531.85 | 22.7 | 207 | 2.50 |
| 538.45 | 23.3 | 203 | 2.45 |
| 553.30 | 24.6 | 204 | 1.89 |
| 558.25 | 25.1 | 202 | 2.60 |
| 570.35 | 26.3 | 201 | 1.80 |





5 **Table 2.** Ion concentrations of bag mean and discrete samples. Unit is µeq L-1. The constant value of 0.30 for the ion-deduced $NaCl/Na_2SO_4$ ratio though the Holocene and late transition occurs because we assume that 23% of $Na^+$ is present as NaCl when $SO_4^{2-}$ concentration exceeds the $Na^+$ concentration. See details of the method in section 2.3.

| | | | Bag mean | | | | | | Top 5 cm | | | | | |
|---|---|---|---|---|---|---|---|---|---|---|---|---|---|---|
| Top sample depth (m) | Age (ka BP1950) | sublimation $NaCl/Na_2SO_4$ | $Cl^-$ | $SO_4^{2-}$ | $Na^+$ | $Ca^{2+}$ | $Cl^-/Na^+$ | Ion-deduced $NaCl/Na_2SO_4$ | $Cl^-$ | $SO_4^{2-}$ | $Na^+$ | $Ca^{2+}$ | $Cl^-/Na^+$ | Ion-deduced $NaCl/Na_2SO_4$ |
| 222.8 | 6.8 | 0.01 | 0.45 | 1.79 | 0.94 | 0.06 | 0.47 | 0.3 | 0.21 | 1.35 | 0.57 | 0.03 | 0.37 | 0.3 |
| 233.8 | 7.2 | 0.02 | 0.36 | 1.76 | 0.88 | 0.08 | 0.41 | 0.3 | 0.29 | 1.75 | 0.92 | 0.06 | 0.32 | 0.3 |
| 241.5 | 7.5 | 0.47 | 0.45 | 1.7 | 0.97 | 0.19 | 0.46 | 0.3 | 0.33 | 1.96 | 1.04 | 0.13 | 0.32 | 0.3 |
| 251.4 | 7.9 | 0.43 | 0.39 | 2.62 | 0.86 | 0.1 | 0.46 | 0.3 | 0.4 | 6.38 | 0.98 | 0.02 | 0.4 | 0.3 |
| 276.7 | 8.9 | 0.41 | 0.66 | 1.87 | 0.96 | 0.23 | 0.69 | 0.3 | 0.32 | 1.6 | 0.62 | 0.17 | 0.51 | 0.3 |
| 287.1 | 9.3 | 0.45 | 0.68 | 1.83 | 0.89 | 0.38 | 0.76 | 0.3 | 0.33 | 2.74 | 0.77 | 3.33 | 0.43 | - |
| 296.5 | 9.6 | 0.37 | 1.24 | 1.9 | 0.92 | 0.15 | 1.35 | 0.3 | 0.89 | 1.67 | 0.53 | 0.12 | 1.69 | 0.3 |
| 303.1 | 9.9 | 0.11 | 1.12 | 1.77 | 0.62 | 0.17 | 1.79 | 0.3 | 0.99 | 1.28 | 0.41 | 0.1 | 2.44 | 0.3 |
| 313.55 | 10.2 | 0.01 | 0.59 | 2.19 | 0.85 | 0.1 | 0.69 | 0.3 | 0.47 | 1.71 | 0.55 | 0.07 | 0.86 | 0.3 |
| 321.8 | 10.5 | 0.15 | 0.61 | 2.56 | 0.75 | 0.17 | 0.81 | 0.3 | 0.63 | 2.26 | 0.73 | 0.05 | 0.87 | 0.3 |
| 334.95 | 10.9 | 0.05 | 0.55 | 1.6 | 0.58 | 0.04 | 0.96 | 0.3 | 0.59 | 1.48 | 0.59 | 0.03 | 1 | 0.3 |
| 350.35 | 11.5 | 0.12 | 0.91 | 1.97 | 0.69 | 0.12 | 1.33 | 0.3 | 0.95 | 2.2 | 0.85 | 0.5 | 1.12 | 0.3 |
| 368.5 | 12.1 | 0.03 | 0.37 | 1.79 | 1.05 | 0.15 | 0.36 | 0.3 | 0.31 | 2.02 | 1.09 | 0.11 | 0.28 | 0.3 |
| 377.3 | 12.5 | 0.02 | 0.26 | 2.06 | 1.19 | 0.11 | 0.22 | 0.3 | 0.43 | 2.64 | 1.85 | 0.17 | 0.23 | 0.3 |
| 386.65 | 13 | 0.24 | 0.46 | 2.51 | 1.68 | 0.17 | 0.27 | 0.3 | 0.29 | 1.77 | 0.92 | 0.1 | 0.32 | 0.3 |
| 395.45 | 13.5 | 0.18 | 0.59 | 2.59 | 1.85 | 0.22 | 0.32 | 0.3 | 0.82 | 3.32 | 2.39 | 0.22 | 0.34 | 0.3 |
| 403.15 | 13.9 | 0.15 | 0.33 | 2.67 | 1.8 | 0.08 | 0.18 | 0.3 | 0.29 | 4.52 | 2.49 | 0.11 | 0.12 | 0.3 |
| 422.95 | 14.9 | 0.26 | 0.45 | 2.56 | 1.6 | 0.15 | 0.28 | 0.3 | 0.54 | 2.24 | 1.66 | 0.25 | 0.33 | 0.3 |
| 430.65 | 15.3 | 0.04 | 0.93 | 2.7 | 2.26 | 0.44 | 0.41 | 0.3 | 0.37 | 2.86 | 1.66 | 0.46 | 0.22 | 0.3 |
| 449.35 | 16.4 | 0.23 | 2.95 | 2.86 | 2.25 | 0.5 | 1.31 | 0.3 | 2.78 | 2.62 | 1.98 | 0.49 | 1.41 | 0.3 |
| 458.15 | 16.9 | 0.22 | 2.07 | 3.08 | 2.85 | 0.54 | 0.73 | 0.12 | 2.68 | 2.51 | 2.98 | 0.46 | 0.9 | 0.45 |
| 465.85 | 17.5 | 0.26 | 3.35 | 3.53 | 3.52 | 1.11 | 0.95 | 0.46 | 3.98 | 4.48 | 4.24 | 1.16 | 0.94 | 0.28 |
| 481.25 | 18.7 | 1.56 | 5.67 | 4.12 | 4.98 | 2.79 | 1.14 | 2.75 | 5.9 | 5.16 | 5.7 | 2.81 | 1.03 | 1.42 |
| 505.45 | 20.6 | 1.02 | 5.64 | 4.3 | 5.41 | 2.46 | 1.04 | 1.95 | 5.01 | 4.5 | 4.74 | 2.59 | 1.06 | 1.49 |
| 521.95 | 21.9 | 0.37 | 5.25 | 4.13 | 4.05 | 1.34 | 1.3 | 0.45 | 4.92 | 4.01 | 3.73 | 1.43 | 1.32 | 0.45 |
| 531.85 | 22.7 | 1.15 | 5.25 | 3.77 | 4.85 | 2.03 | 1.08 | 1.8 | 5.06 | 3.93 | 4.53 | 1.8 | 1.12 | 1.13 |
| 538.45 | 23.3 | 0.11 | 4.3 | 3.51 | 3.62 | 1.33 | 1.19 | 0.67 | 4.48 | 3.43 | 3.63 | 1.44 | 1.23 | 0.83 |
| 553.3 | 24.6 | 0.03 | 4.73 | 5.92 | 4.87 | 2.47 | 0.97 | 0.41 | 2.18 | 10.37 | 4.51 | 2.03 | 0.48 | 0 |
| 558.25 | 25.1 | 3.2 | 5.69 | 4.62 | 5.17 | 2.59 | 1.1 | 1.55 | 6.71 | 4.6 | 6.08 | 3.37 | 1.1 | 3.95 |
| 570.5 | 26.3 | 0 | 4.87 | 4.54 | 4.75 | 2.59 | 1.02 | 1.44 | 3.98 | 3.9 | 3.29 | 2.45 | 1.21 | 1.28 |



**Table 3.** Average ion and salt concentrations of the LGM and early Holocene in the Dome C and Dome Fuji ice cores.

|  |  | Age (ka BP1950) | $Ca^{2+}$ | $Na^+$ | $SO_4^{2-}$ | $CaSO_4$ | $Na_2SO_4$ |
|---|---|---|---|---|---|---|---|
| Dome C | LGM | 18–25 | 1.18 | 4.41 | 2.21 | 1.18 | 1.01 |
|  | Early Holocene | 9 | 0.03 | 0.68 | 0.95 | 0.03 | 0.26 |
| Dome Fuji | LGM | 18–25 | 1.55 | 4.41 | 2.27 | 1.55 | 0.81 |
|  | Early Holocene | 9 | 0.12 | 0.54 | 1.09 | 0.07 | 0.32 |



**Figures**

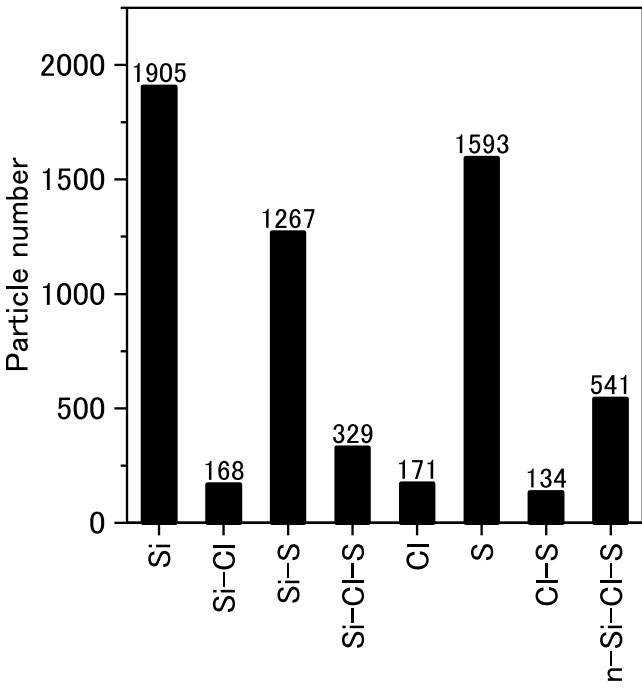

**Fig. 1.** Number distribution of Si, Cl, and S particles from all samples, based on sublimation analyses. The bars depict combinations of Si-, S-, and Cl-containing particles. For example, particles containing Si but not S and Cl are categorized in the Si bar. This bar indicates silicate dust. The Si-Cl bar indicates a mixture of chloride salts and silicate dust. The Si-S bar indicates a mixture of sulfate salts and silicate dust. The Si-Cl-S bar indicates a mixture of chloride salts, sulfate salts, and silicate dust. The Cl bar indicates chloride salts. The S bar indicates sulfate salts. The Cl-S bar indicates a mixture of chloride and sulfate salts. The n-Si-Cl-S bar indicates particles other than chloride salts, sulfate salts, and silicate dust.



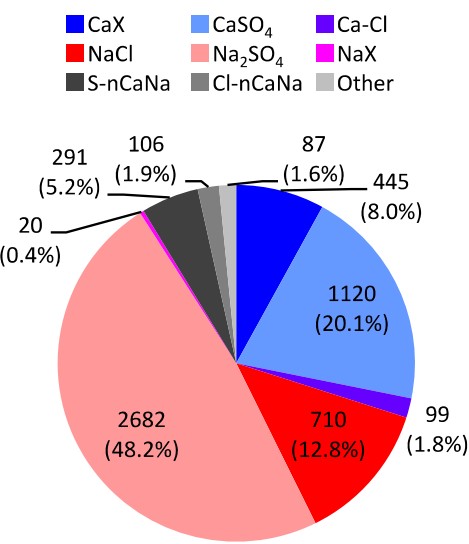

**Fig. 2.** Number fraction of soluble components of all particles. CaX indicates particles containing Ca but neither S nor Cl. Ca-Cl indicates particles containing Ca and Cl. Since the eutectic temperature of $CaCl_2$ is low (approximately -50°C), it is

15    not certain that such salt can be measured by the sublimation method. Therefore, we call particles containing Ca and Cl as Ca-Cl rather than $CaCl_2$. NaX indicates particles containing Na but no Si, S, or Cl. S-nCaNa indicates sulfate salts other than $Na_2SO_4$ and $CaSO_4$. Cl-nCaNa indicates chloride salts other than NaCl and Ca-Cl particles. Other indicates soluble salts other than sulfate and chloride salts. These nine categories cover all soluble salt particles.



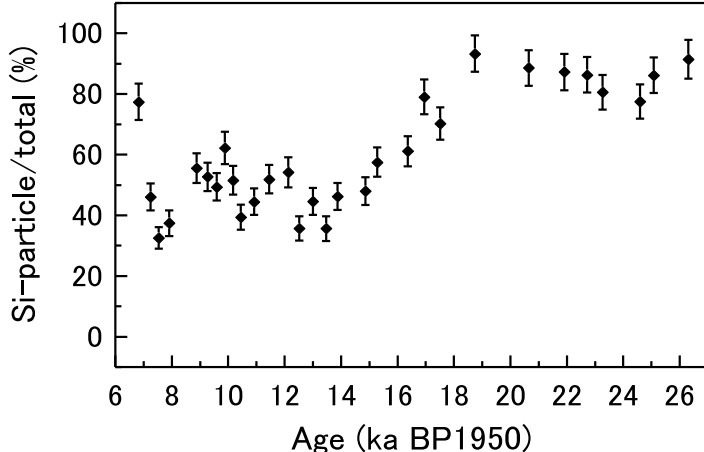

10 **Fig. 3.** Number ratio of Si-containing particles to total non-volatile particles.





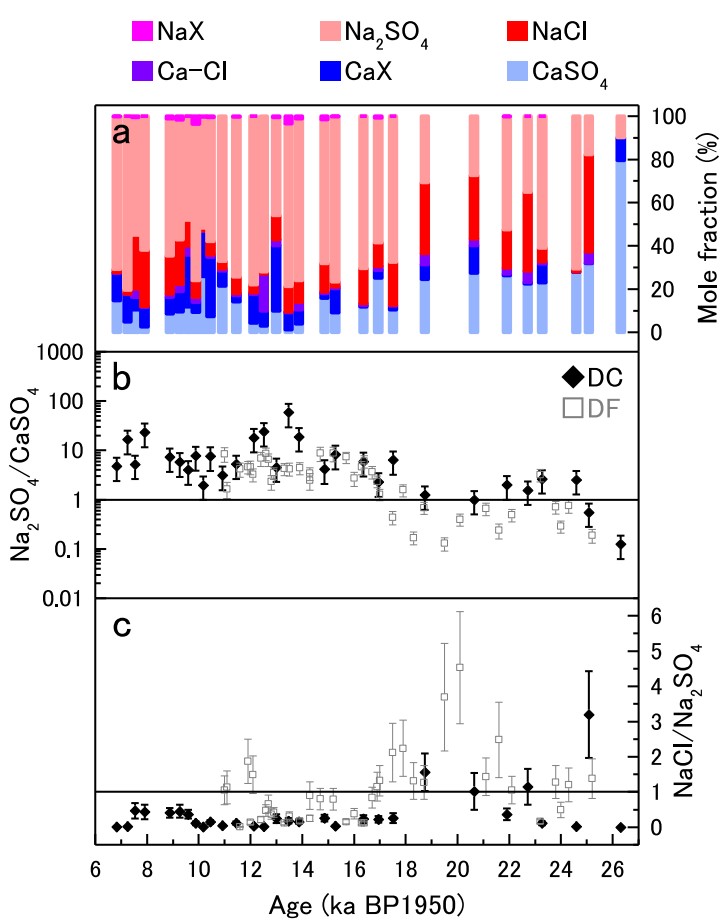

**Fig. 4.** Time series variation of Na-salt and Ca-salt ratios. (a) Mole fraction of Na-salts and Ca-salts. (b) Mole ratio of $Na_2SO_4$ to $CaSO_4$. (c) Mole ratio of NaCl to $Na_2SO_4$. Dome Fuji records in (b) and (c) from Oyabu et al. (2014) are given on the AICC2012 age scale using tie points in Fujita et al. (2015).





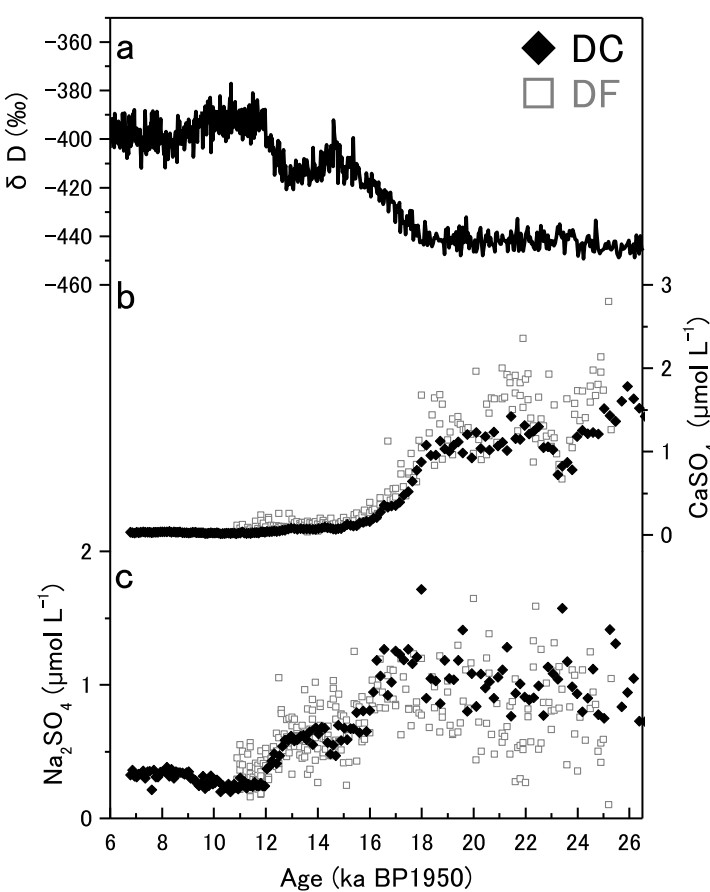

**Fig. 5.** Time series of the concentrations. Black symbols and symbols indicate data from Dome C and grey symbols indicate data from Dome Fuji (Oyabu et al., 2014). Dome Fuji records are given on the AICC2012 age scale. (a) δD record of Dome C ice core from Jouzel et al. (2007). (b) $CaSO_4$ concentration. (c) $Na_2SO_4$ concentration

