# Peer review of "Chemical composition of soluble and insoluble particles around the last termination preserved in the Dome C ice core, inland Antarctica"

_Climate of the Past, 2016_

## Referee Comment (RC1) · Anonymous Referee #1 · 6 May 2016

In this manuscript the authors compare the chemical composition and concentrations of non-volatil aerosols from the EPICA Dome C and the Dome Fuji ice core. They use SEM and EDS analysis to estimate the concentrations of the main soluble salt particles from 6.8 kaBP to 25 kaBP. They then infer how the ratios changed during the termination to identify two different regimes of transport between the two periods.

The manuscript is generally poorly written and lacks scientific rigor, particularly in the statistical analysis of the data. The main weakness of this manuscript is that the authors have very few data points and make many statements based on one or two data points that may or may not be outliers and/or statistically significant. As an example, in chapter 3.3, page 7, lines 12-18 the authors compare their analyzed top 5 cm with

the bag means and select one section to discuss where the top 5 cm does not agree with the bag mean and to justify a low ratio of NaCl/Na2SO4. However they completely disregard other LGM samples that also have low LGM NaCl/Na2SO4 ratios but where the top 5 cm agrees with the bag mean. As another example the authors suggest in chapter 3.4, page 8, line 25 that the there is a 600 year shift between the maximal sulfatization at Dome C and DF. Such a shift is completely unsupported by the data in Figure 4c. There are many other examples. These unsupported claims, combined with statistical mistakes like giving averages and standard deviations of clearly non-Gaussian data or identifying maxima/minima to a precision of 100 years with 30 data points in a period of 20,000 years only underline the flawed analysis of the dataset. I do not doubt the profound chemical knowledge of the authors, but I cannot believe their conclusions based on such poor analysis of the data. Finally, although the salt ratios have never been measured before, there is nothing new about the conclusions of this paper. For these reasons I suggest to reject this manuscript.

---

## Referee Comment (RC2) · Anonymous Referee #2 · 27 May 2016

The paper is focused on the study of the chemical composition of soluble and soluble particles in the period 6.8 – 26.3 kyr BP in Dome C ice core by using EDS analysis upon sample sublimation and on the comparison of the achieved results with Dome Fuji ice core. This paper strongly resembles in outline a previous paper dealing with Dome Fuji ice core by the same leading Author (Oyabu et al., 2014, J. Geophys. Res.) which is mentioned often in the present paper but actually adjusted to Dome C ice core both as text and as figures (for instance, figures 2, 3, 4 in this paper are just specular to figures 4a, 3, 4b of Oyabu et al, 2014). The paper is filled with very general and hardly meaningful or poorly referenced statements (see examples below) and, although SEM-EDS compositional data from EDC had never been presented before, they do not add

significant information to the current knowledge of the chemical composition of the soluble and insoluble particulate deposited onto the East Antarctic plateau during the Last Termination. Moreover, the scientific design is not described clearly and also some methodological choices (used data sets for comparison with sublimation-EDS method, number of data acquired by the latter method, etc.), do not appear suitably supported (see detailed comments below). Finally, the drawn conclusions are whether already well known from similar and ion composition and dust measurements (f.i. when reporting the main soluble salts as CaSO4, Na2SO4 and NaCl and a lower fraction of NaCl during the LGM or when inferring two different transport patterns during the Last Termination, respectively) or scarcely meaningful (invoking both a "uniform" marine environment in the sectors affecting Dome C and Dome Fuji sites or that both sites are influenced by a large part of the Southern Ocean). Hence, for all these reasons, I believe that unfortunately the paper cannot be accepted for publication on "Climate of the Past" journal.

Here below some specific remarks are reported as examples but others can be found along the text.

Abstract Page 1, lines 21-23. "However. . . . . .Ca2+". Such a conclusion could be easily drawn by comparing already published sulphate, sodium and calcium records from Dome C and Dome Fuji ice cores. It certainly does not take a time-consuming and detailed EDS analysis to claim this. The fact that the EDS analysis on single particles in the end lead to already well known facts is one of the main flaws of the paper.

Introduction Page 2, lines 26-31. This is an example of a mix of uncorrect and obvious statements. The different amount of Na2SO4 found in Talos Dome and Dome Fuji cannot tell by itself on the different yield of reaction R2 (line 22) since it depends also on the actual total amount of each components deposited at the two sites and this is likely to be discriminant for determining the final content of sodium sulphate particles. Moreover, it is way too simplistic and uncorrect justifying the claimed different sulfatization contribution of NaCl at the two sites by the supposed larger time needed to

reach the plateau. The fact that Dome Fuji is about 1000 km inland and Talos Dome around 250 km far from the coastline does not mean that marine air masses necessarily take a shorter time in reaching the latter since it depends on transport routes (not necessarily and often not in straight line!) and on the speed of air masses, in turn depending on zonal/regional atmospheric circulation, following different patterns in the different climatic regimes. Finally, R2 is supposed to occur in the atmosphere, so what do "post-depositional effects" (line 29) have to do with it?

Methods The sample data set (n. 30 samples as a whole) is not large enough, considering that they span periods with dramatic differences in load and chemical composition, to assure a statistical significance.

This chapter shows many unclear and uncorrect statements and descriptions. Here below a few examples. Page 3, line 20. "pulverized on of the faces...range" is perfectly unclear. Page 3, line 31. Acceleration voltage is in kV, not keV. Page 3 line 31 – page 4, line 1. The sentence is not clear at all. Maybe it is supposed to be explained by lines 23-24 in the same page but there is no clear information on the approach used to determine the particle diameter (measured or "equivalent diameter" calculated on the basis of the section area?) and error. Paragraph 2.3. I see no point in taking two different dat sets for $Ca^{2+}$, $Na^+$ and $SO_4^{2-}$ (Ion Chromatography) and $Cl^-$ and $NO_3^-$ (Continuous Flow Analysis) while the same parameters were also available from the Ion Chromatography measurements on the same samples.

---

## Referee Comment (RC3) · Anonymous Referee #3 · 31 May 2016

In this paper, Oyabu and coauthors investigate the chemical composition of nonvolatile (soluble and insoluble) particles at Dome C during the last glacial maximum, the last termination and Holocene (26.3ka to 6.8ka B.P.). The authors are undoubtedly among the maximum experts in the field. They present for the first time SEM-EDS compositional data from Dome C on a set of 30 samples. Because this type of measurements is extremely time-consuming, I think 30 samples is a reasonable number for a publication. The methodology followed in this work is the same as in Oyabu et al., 2014 (DOI: 10.1002/2014JD022030), thus data from the two ice cores drilled in different sectors of East Antarctica are fully comparable. This is an advantage.

While this kind of data is potentially very interesting, I found the discussion part rather

weak: some statements are not suitably supported, and the paper in general does not add much to what is already known from Dome Fuji.

Therefore, I encourage the authors to re-structure the discussion part of this work including some major revisions.

Some detailed points:

Pg4, lines 20-25: the method used to determine particle size (although size is not discussed further in the ms) is not clear at all. If you regard each particle as a sphere of revolution around the major axis, you do not consider the dramatic difference between aspect ratios from top orientation and side orientation that is typical of clay minerals that we expect to find at DC. However, if I understand well, you overcome this problem by measuring the shadow area on a subset of particles selected at random, is this correct? Please explain your method in detail and add a reference.

If particle size determination with this method has been carried out on Dome C particles in the framework of this work, why you do not discuss in the paper the relation between particle size and composition, in order to highlight if there are some soluble salts that are clearly smaller/larger than others and/or give a size range to these soluble aerosols?

Pg6, lines 7-8: to support the statement "the ratio of insoluble dust to non-volatile aerosols appears to change with dust concentration" can you compare this ratio to dust concentration data from independent measurements (Lambert et al., 2008)?

Pg6, lines 31-32: I hardly agree that the pattern of changes during the last termination is similar between DC and DF. Actually, looking at fig. 4b, for example, it seems that only for major glacial-interglacial changes the two sites are in agreement, while (despite the few data) some differences arise during ACR and the minimum preceding the Holocene. It is surprising to see at Dome C a higher $Na_2SO_4/CaSO_4$ ratio during ACR and a lower ratio during the dust minimum. The pattern of Dome Fuji in this sense

seems much closer to what is expected. Can you comment on this difference?

Pg7, line 30: here you suggest a post-depositional process for Na2SO4 formation. I admit I am lost because the (pre-depositional and/or post-depositional) processes involved in the formation of Ca and Na salts are not clear at all. CaCO3 is believed to react with SO4 in the atmosphere, whereas sulfatisation of NaCl is a post-depositional process? Or both?

Pg.8 lines 25-26: I think it is a risk with such few data to state that most NaCl sulfatized earlier at Dome C than at Dome F.

I am confident that a deep re-writing of this paper will lead to substantial improvements.

―――――――――――――――――――

---

## Author Comment (AC1) · 30 Jun 2016

(Original review comments are in *Italic letters*, and our replies are in blue color.)

Dear Editor, Dr. Barbara Stenni

We thank all the reviewers for taking the time to read our paper. In this study, we measured soluble salt compositions of Dome C and compared the result with salt compositions of Dome Fuji. We newly found that 1) changes in salt composition of Dome C with glacial-interglacial time scale is similar to that of Dome Fuji, 2) more $Na_2SO_4$ presented during LGM than what previous studies expected, and 3) greater NaCl sulfatization occurred in Dome C compared to Dome Fuji.

Reviewers 1 and 2 have as their principle issues with the paper the same two points and we should address these first as they address the philosophy of the paper. Firstly they both object that we have only 30 samples covering 20 ka. Of course we agree we would like to have many more samples analyzed, but this type of analysis, involving a long sublimation step and visualizing individual particles, is very time consuming, and took the lead author a full year of work to obtain. We agree that we should be more cautious in interpreting our data at too fine a time resolution, and will correct that if we are allowed to submit a revised version. However the data are novel, and allow discussion at the millennial scale, which is already interesting for the transition from the last glacial to the Holocene.

The second objection is that we do not learn much that is new compared to bulk chemical analysis and compared to what was already done at Dome Fuji. It is true that one can successfully deduce many of the conclusions from bulk analysis, making particular assumptions. However it is only by observing individual salt particles that one can show that those assumptions were correct. The point about Dome C compared to Dome Fuji does not seem reasonable to us. It is only by comparing data from two sites that one can tease out whether they show the same of a different timing and result. Even if the second site just confirms what was found at the first one, is this a reason not to publish it? If so, there is a huge amount of the ice core literature that should never have been published.

Beyond these two points, reviewers 1 and 2 make some general criticisms of our analysis but only provide a few examples. These would each easily be rectified in a revision, and in at least one case highlighted by reviewer 1, they arise from a misunderstanding of what we are saying. In general we agree that our discussion could be clearer and more focused. We hope we have a chance to submit a revised manuscript. We deal with the comments point by point below.

Sincerely yours,
Ikumi Oyabu
On behalf of the co-authors

National Institute of Polar Research
10-3 Midorichou, Tachikawa, Tokyo 190-8518, Japan
TEL: +81-42-512-0760

*Referee #1*

*In this manuscript the authors compare the chemical composition and concentrations of non-volatil aerosols from the EPICA Dome C and the Dome Fuji ice core. They use SEM and EDS analysis to estimate the concentrations of the main soluble salt particles from 6.8 kaBP to 25 kaBP. They then infer how the ratios changed during the termination to identify two different regimes of transport between the two periods.*

Thank you for your reviewing. In this study, we measured soluble salt compositions of Dome C and compared the result with salt compositions of Dome Fuji. We newly found that 1) changes in salt composition of Dome C with glacial-interglacial time scale is similar to that of Dome Fuji, 2) more $Na_2SO_4$ presented during LGM than what previous studies expected, and 3) greater NaCl sulfatization occurred in Dome C compared to Dome Fuji. Many studies have deduced salt compositions from bulk ion analysis, however, only individual particle analysis can confirm if their assumptions were correct or not. We believe the data and three major findings are novel, and worth a publication.

*The manuscript is generally poorly written and lacks scientific rigor, particularly in the statistical analysis of the data. The main weakness of this manuscript is that the authors have very few data points and make many statements based on one or two data points that may or may not be outliers and/or statistically significant.*

We agree that we should not make a centennial time scale discussion from 30 data points. We should take more care with the limitation of time resolution and significance of each data point. On the other hand, we would ask for your understanding that the sublimation method, involving a long sublimation step and visualizing individual particles, is extremely time consuming. In total, it took more than 1 year to obtain 30 data points.

*As an example, in chapter 3.3, page 7, lines 12-18 the authors compare their analyzed top 5 cm with the bag means and select one section to discuss where the top 5 cm does not agree with the bag mean and to justify a low ratio of NaCl/Na2SO4. However they completely disregard other LGM samples that also have low LGM NaCl/Na2SO4 ratios but where the top 5 cm agrees with the bag mean.*

Unfortunately it seems that you misunderstood what we wrote in section 3.3. page 7, lines 12-18 and we will have to word it more carefully. We do not at all disregard the samples where the bag mean agrees with the top 5 cm. What we are doing is trying to understand the variability of the $NaCl/Na_2SO_4$ ratio. The sublimation was carried out using the top ~5 cm of a 55 cm ice length. We therefore consider first whether variability simply arises from the fact that the top 5 cm is not representative of the ice around it. This seems to be the case in one instance (24.6 ka) but not the others so we have to seek another explanation for the variability.

*As another example the authors suggest in chapter 3.4, page 8, line 25 that the there is a 600 year shift between the maximal sulfatization at Dome C and DF. Such a shift is completely unsupported by the data in Figure 4c. There are many other examples. These unsupported claims, combined with statistical mistakes like giving averages and standard deviations of clearly non-Gaussian data or identifying maxima/minima to a precision of 100 years with 30 data points in a period of 20,000 years only underline the flawed analysis of the dataset. I do not doubt the profound chemical knowledge of the authors, but I cannot believe their conclusions based on such poor analysis of the data.*

We agree that we cannot discuss centennial time scales as we briefly did in chapter 3.4, page 8, line 25. We should reword this part. On the other hand, we can discuss salt compositions with each data point. For example, there is no $NaCl/Na_2SO_4$ ratio (Fig. 4c) exceeding 1 after 17.5ka, while 4 points show a higher ratio in the earlier period. These data mean that most of $Na^+$ was $Na_2SO_4$ after 17.5 ka, whereas there is more variability in the Na-salt present before 18.7ka. This large variability in LGM is different from what previous studies led us to expect. If we have a chance to revise, we would like to discuss each data point in this way.

*Finally, although the salt ratios have never been measured before, there is nothing new about the conclusions of this paper. For these reasons I suggest to reject this manuscript.*

Although, glacial-interglacial time scale salt composition is similar between Dome C and Dome Fuji, as we wrote in the top of this letter, we have new findings. Actually, the similarity of salt composition is also a new finding, since the literature about the ionic chemistry of Dome Fuji is quite limited, and it was not compared to Dome C before. We do not think that new data from a site thousands of km from sites where similar methods were previously used can be dismissed as "nothing", and it does not seem to us a good argument that new data which agree with older data from a completely different site are not worth publishing.

*Refree #2*

*The paper is focused on the study of the chemical composition of soluble and soluble particles in the period 6.8 – 26.3 kyr BP in Dome C ice core by using EDS analysis upon sample sublimation and on the comparison of the achieved results with Dome Fuji ice core. This paper strongly resembles in outline a previous paper dealing with Dome Fuji ice core by the same leading Author (Oyabu et al., 2014, J. Geophys. Res.) which is mentioned often in the present paper but actually adjusted to Dome C ice core both as text and as figures (for instance, figures 2, 3, 4 in this paper are just specular to figures 4a, 3, 4b of Oyabu et al, 2014).*

Thank you for reviewing.

In order to compare the Dome C result with Oyabu et al. 2014, we did make similar figures and discussion. Since no-one else does this type of analysis, we wanted to directly compare between Dome Fuji and Dome C. There are many papers that put data on the same figure, and discuss how the two data sets are different/similar, and why. We are not sure if the reviewer is criticizing us for using this common method, but we don't see why we would not be allowed to follow this way of comparing data from different sites, as others have done?

*The paper is filled with very general and hardly meaningful or poorly referenced statements (see examples below) and, although SEM- EDS compositional data from EDC had never been presented before, they do not add significant information to the current knowledge of the chemical composition of the soluble and insoluble particulate deposited onto the East Antarctic plateau during the Last Termination.*

*Moreover, the scientific design is not described clearly and also some methodological choices (used data sets for comparison with sublimation-EDS method, number of data acquired by the latter method, etc.), do not appear suitably supported (see detailed comments below).*

*Finally, the drawn conclusions are whether already well known from similar and ion composition and dust measurements (f.i. when reporting the main soluble salts as CaSO4, Na2SO4 and NaCl and a lower fraction of NaCl during the LGM or when inferring two different transport patterns during the Last Termination, respectively) or scarcely meaningful (invoking both a "uniform" marine environment in the sectors affecting Dome C and Dome Fuji sites or that both sites are influenced by a large part of the Southern Ocean). Hence, for all these reasons, I believe that unfortunately the paper cannot be accepted for publication on "Climate of the Past" journal.*

In this paper, we measured soluble salt compositions of Dome C and compared the result with salt compositions of Dome Fuji. We newly found that 1) changes in salt composition of Dome C with glacial-interglacial time scale is similar to that of Dome Fuji, 2) more $Na_2SO_4$ presented during LGM than what previous studies expected, and 3) greater NaCl sulfatization occurred in Dome C compared to Dome Fuji. Many studies have deduced salt compositions from bulk ion analysis, however, only individual particle analysis can confirm if their assumptions were correct or not. We believe data and three major findings are

novel, and worth a publication. We accept that especially the statement about the Southern Ocean in the last paragraph of the conclusion is too vague and should be revised. We would like to emphasize what we have found if we are allowed to revise.

Details are answered point by point as below.

*Here below some specific remarks are reported as examples but others can be found along the text.*

*Abstract Page 1, lines 21-23. "However. . . . . .Ca2+". Such a conclusion could be easily drawn by comparing already published sulphate, sodium and calcium records from Dome C and Dome Fuji ice cores. It certainly does not take a time-consuming and detailed EDS analysis to claim this. The fact that the EDS analysis on single particles in the end lead to already well known facts is one of the main flaws of the paper.*

For the first, $Na^+$, $Ca^{2+}$ and $SO_4^{2-}$ concentration of Dome Fuji during the last termination has not been published yet (Only their flux has been published by Oyabu et al., 2014).

In this study, we found that sulfatization of NaCl occurred more at Dome C than Dome Fuji for the first time. To explain this reason, we compared ion concentrations, and found the reason that lower $Ca^{2+}$ concentration of Dome C allowed NaCl to contact with more $H_2SO_4$ compared to Dome Fuji. If you have Dome Fuji dataset and compare with Dome C, of course you can draw a conclusion "$Ca^{2+}$ concentration of Dome Fuji is higher than that of Dome C". However, can you draw the same conclusion with us from only ion concentration? We think you can deduce salt compositions from ion concentration, but you never known real salt compositions without individual particle analysis. We do not agree at all your comment.

*Introduction Page 2, lines 26-31. This is an example of a mix of uncorrect and obvious statements. The different amount of Na2SO4 found in Talos Dome and Dome Fuji cannot tell by itself on the different yield of reaction R2 (line 22) since it depends also on the actual total amount of each components deposited at the two sites and this is likely to be discriminant for determining the final content of sodium sulphate particles.*

With regard to "The different amount of Na2SO4・・・・", we agree with you. It is important to consider not only yield of reaction but also total amount. However, Iizuka et al. 2013 discussed not absolute concentration but relative ratio. They found that $NaCl/Na_2SO_4$ is higher in Talos Dome than in Dome Fuji, despite similar ionic balance ($Cl^-$ and $SO_4^{2-}$ concentrations exceeded $Na^+$ concentration). This evidence suggests that the NaCl of Talos Dome had less contact with $H_2SO_4$ than particles reaching to Dome Fuji. To explain this result, they focused on the different yield of reaction R2 and post depositional process.

*Moreover, it is way too simplistic and uncorrect justifying the claimed different sulfatization contribution of NaCl at the two sites by the supposed larger time needed to reach the plateau. The fact that Dome Fuji is about 1000 km inland and Talos Dome around 250 km far from the coastline does not mean that marine air masses necessarily take a shorter time in reaching the latter since it depends on transport routes (not necessarily and often not in straight line!) and on the speed of air masses, in turn depending on zonal/regional atmospheric circulation, following different patterns in the different climatic regimes.*

As you comment, marine air masses do not necessarily take a shorter time in reaching Talos Dome. When strong low pressure occurs, marine air mass reaches to 1000 km inland in a very short time. In such case, NaCl is probably transported without sulfatization. However, Iizuka et al. 2013 were not observing such short time scale event, rather averaged salt compositions in the 4 climatic stages. It is certainly true that the average transportation time or residence time of sea-salt is probably longer for the salt reaching to Dome Fuji than that for the salt reaching to Talos Dome.

*Finally, R2 is supposed to occur in the atmosphere, so what do "post-depositional effects" (line 29) have to do with it?*

Reaction R2 can occur not only in the atmosphere but also in the snow (e.g., Röthlisberger et al., 2003, Iizuka et al., 2004). Iizuka et al. 2013 considered that a higher accumulation rate prevents NaCl from reacting with $H_2SO_4$ in the snow. The higher accumulation rate at Talos Dome leads to less sublimation of snow and acid at the surface snow than that at Dome Fuji. Less snow sublimation means that less $H_2SO_4$ and NaCl migrate from the inside to the surface of the snow crystals, thus providing less chance of contact between the acid and salt.

*Methods*

*The sample data set (n. 30 samples as a whole) is not large enough, considering that they span periods with dramatic differences in load and chemical composition, to assure a statistical significance.*

We would like to ask for your understanding that the sublimation method, involving a long sublimation step and visualizing individual particles, is extremely time consuming. In total, it took more than 1 year to obtain 30 data points. We agree that we cannot discuss with centennial time scale, but 30 data allow discussion at the millennial scale, which is already interesting for the transition from the last glacial to the Holocene. Therefore, we believe 30 samples is a reasonable number for a publication.

*This chapter shows many unclear and uncorrect statements and descriptions. Here below a few examples. Page 3, line 20. "pulverized on of the faces. . .range" is perfectly unclear. Page 3, line 31. Acceleration voltage is in kV, not keV. Page 3 line 31 – page 4, line 1. The sentence is not clear at all. Maybe it is supposed to be explained by lines 23-24 in the same page but there is no clear information on the approach used to determine the particle diameter (measured or "equivalent diameter" calculated on the basis of the section area?) and error. Paragraph 2.3. I see no point in taking two different dat sets for Ca2+, Na+ and SO42- (Ion Chromatography) and Cl- and NO3- (Continuous Flow Analysis) while the same parameters were also available from the Ion Chromatography measurements on the same samples.*

Page 3, line 20:

Each ice sample size is 10*4*3 cm³, but we do not need the whole volume. To put the sample in the sublimation chamber, we prepared 1g pulverized ice by shaving the sample surface. Typically we shaved a section 5-7 cm long in the depth dimension. We can clearly make this clearer if the reviewer finds it unclear.

kV – OK

Page 3, line 31 – page 4, line1:

This sentence mentioned how we took data. When elemental compositions are analyzed by EDS, atomic ratio (%) and the error (%) are simultaneously calculated. We decided to take data only when atomic ratio is more than twice as high as the error in order to avoid to take a particle data containing too little of target element.

Lines 23-24 is written about error of the spherical approximation and reproducibility. We will write more in detail if we are allowed to revise. The reason why we measured particle size is to calculate particle volume to obtain its mass. We measured longest diameter of section area of each particle (R) and calculated particle volume (V) by assuming that all particle is idealized sphere. $V = 4/3\pi(R/2)^3$. To quantify the error of this spherical approximation, we measured section area of 200 particles by graphics software, and then compared circle shadow area ($A = \pi(R/2)^2$). As a result of the comparison between actual shadow area and circle shadow area, 95% of the particles had areas that differed by less than 20%. So we took 20% as spherical approximation.

"The analysis error of SEM-EDS" in line 24 means reproducibility (coefficient of variation, or CV) of the EDS analysis. A particle was selected at random. If, for example, it contained Na and S, then we repeatedly (20×) measured its atomic ratios of Na. Then the ratio of the standard deviation to the average value of the 20 measurements ($CV_{Na}$) was calculated. CV values of the six major elements (Na, Mg, Ca, K, S, and Cl) were obtained in the same way. The average coefficient of variation of these elements was similar to each other, so we used their average value (CV = 22%).

Paragraph 2.3:

$Cl^-$ concentration provided by Röthlisberger et al. (2003) was measured not by CFA but ion chromatography. $NO_3^-$ concentration has been measured by ion chromatography, but is not available online, so we chose to use the CFA dataset that is available. This is justified as there are no major differences between CFA and IC as shown by Littot et al. (2002).

Littot et al., 2002, Comparison of analytical methods used for measuring major ions in the EPICA Dome C (Antarctica) ice core. *Annals of Glaciology, Vol 35*, 35(1), 299–305).

*Referee #3*

*In this paper, Oyabu and coauthors investigate the chemical composition of nonvolatile (soluble and insoluble) particles at Dome C during the last glacial maximum, the last termination and Holocene (26.3ka to 6.8ka B.P.). The authors are undoubtedly among the maximum experts in the field. They present for the first time SEM-EDS compositional data from Dome C on a set of 30 samples. Because this type of measurements is extremely time-consuming, I think 30 samples is a reasonable number for a publication. The methodology followed in this work is the same as in Oyabu et al., 2014 (DOI: 10.1002/2014JD022030), thus data from the two ice cores drilled in different sectors of East Antarctica are fully comparable. This is an advantage.*

*While this kind of data is potentially very interesting, I found the discussion part rather weak: some statements are not suitably supported, and the paper in general does not add much to what is already known from Dome Fuji.*

*Therefore, I encourage the authors to re-structure the discussion part of this work including some major revisions.*

We deeply appreciate your reviewing of our manuscript and encouraging. We are grateful for your understanding the value of our sublimation dataset. We accept your points and would like to revise substantially.

*Some detailed points:*

*Pg4, lines 20-25: the method used to determine particle size (although size is not discussed further in the ms) is not clear at all. If you regard each particle as a sphere of revolution around the major axis, you do not consider the dramatic difference between aspect ratios from top orientation and side orientation that is typical of clay minerals that we expect to find at DC. However, if I understand well, you overcome this problem by measuring the shadow area on a subset of particles selected at random, is this correct? Please explain your method in detail and add a reference.*

The reason why we measured particle size is to calculate particle volume to obtain its mass. We measured longest diameter of shadow area of each particle (R) and calculated particle volume (V) by assuming that all particle is idealized sphere. $V = 4/3\pi(R/2)^3$. To quantify the error of spherical approximation, we measured actual shadow area of extracted particles selected at random by using graphics software, and then compared circle shadow area ($A = \pi(R/2)^2$). SEM image does not provide us 3D information, we do not consider Z axis. As a result of the comparison between actual shadow area and circle shadow area, 95% of the particles had areas that differed by less than 20%.

*If particle size determination with this method has been carried out on Dome C particles in the framework of this work, why you do not discuss in the paper the relation between particle size and composition, in order to highlight if there are some soluble salts that are clearly smaller/larger than others and/or give a size range to these soluble aerosols?*

We examined particle size distribution by focusing on the Holocene (6 samples) and LGM (6 samples). It should be note that we do not collect data of less than 0.8μm diameter due to analytical resolution of our EDS. Also, we do not collect data of particularly large particle (larger than 10μm). We defined soluble, insoluble and mixture of insoluble and soluble particles as follows;

Soluble particle: particle does not contain Si

Insoluble particle: particle containing Si but no S and Cl

Mixed (Soluble + insoluble particle): particle containing Si and S, Si and Cl, or Si, S and Cl.

Histogram and median diameter is shown in Fig. S1. We found that median diameter of soluble particles is significantly smaller than that of insoluble and mixed particles for both LGM and Holocene. On the other hand, there is no significant difference between insoluble and mixed particles for both LGM and Holocene (p=0.907 for LGM and p=0.645 for Holocene).

Also, we compared the size difference between LGM and Holocene, but there is no significant difference as well. This result is not consistent with previous studies such as Delmonte et al. 2004, Climate Dynamics. To find size difference between different climate stages, it may be needed to analyze a wider size range. We would like to add these discussions in the manuscript.

[Figure]

Fig. S1

*Pg6, lines 7-8: to support the statement "the ratio of insoluble dust to non-volatile aerosols appears to change with dust concentration" can you compare this ratio to dust concentration data from independent measurements (Lambert et al., 2008)?*

We plotted dust concentration and Si-particle ratio vs δD in Fig. S2, and found good agreement. This figure indicates that fraction of Si containing particle to solid aerosols decreased along with warming during the last termination. We hope this evidence supports our statement in P6, lines 7-8.

[Figure]

Comparison between dust mass and Si-particle/total ratio during the last termination

Fig. S2

*Pg6, lines 31-32: I hardly agree that the pattern of changes during the last termination is similar between DC and DF. Actually, looking at fig. 4b, for example, it seems that only for major glacial-interglacial changes the two sites are in agreement, while (despite the few data) some differences arise during ACR and the minimum preceding the Holocene. It is surprising to see at Dome C a higher Na2SO4/CaSO4 ratio during ACR and a lower ratio during the dust minimum. The pattern of Dome Fuji in this sense seems much closer to what is expected. Can you comment on this difference?*

It is as you have pointed out. Major glacial-interglacial changes are similar to each other, but details are different. We will revise this part.

We tried tentatively to explain a higher $Na_2SO_4/CaSO_4$ ratio during ACR and a lower ratio during the dust minimum for the Dome C ice. There is a possibility that composition of Ca-salt changed around ACR. However, we are not very confident in our interpretation at this moment. We need to investigate the significance of the higher values of the ratio of $Na_2SO_4/CaSO_4$ during the ACR, and in a new version we would also look at its evolution in the ion-deduced record of Fig. 5.

*Pg7, line 30: here you suggest a post-depositional process for Na2SO4 formation. I admit I am lost because the (pre-depositional and/or post-depositional) processes involved in the formation of Ca and Na salts are not clear at all. CaCO3 is believed to react with SO4 in the atmosphere, whereas sulfatisation of NaCl is a post-depositional process? Or both?*

Reaction $2NaCl + H_2SO_4 \rightarrow Na_2SO_4 + 2HCl$ can occur both in the atmosphere and after deposition. $Na_2SO_4$ was found from not only snow but was also present in atmospheric aerosol (Hara et al., 2014, Atmos. Chem. Phys.). Unfortunately, not only previous studies but also this study do not distinguish where the sulfatization of NaCl occurred. In our discussion, it is reasonable to consider that major $Na_2SO_4$ formation occurred after deposition because $Cl^-/Na^+$ ratio well agrees with bulk seawater ratio. Of course we cannot claim 100% of sulfatization occurred in the snow, if major sulfatization occurred in the atmosphere, $Cl^-/Na^+$ ratio should have more variability.

*Pg.8 lines 25-26: I think it is a risk with such few data to state that most NaCl sulfatized earlier at Dome C than at Dome F.*

Yes, we agree with you. Time resolution of sublimation analysis is about 600 years. We cannot make such a centennial time scale discussion. We will reword this part.

*I am confident that a deep re-writing of this paper will lead to substantial improvements.*

Thank you again for your comments.